

# Revisiting the Agung 1963 volcanic forcing — impact of one or two eruptions

Ulrike Niemeier[1], Claudia Timmreck[1], and Kirstin Krüger[2]

[1]Max Planck Institute for Meteorology, Bundesstr. 53, 20146 Hamburg, Germany
[2]Department of Geosciences, University of Oslo, Blindern, 0315 Oslo, Norway

**Correspondence:** U. Niemeier (ulrike.niemeier@mpimet.mpg.de)

**Abstract.** In 1963 a series of eruptions of Mt. Agung, Indonesia, resulted in the 3rd largest eruption of the 20th century and claimed about 1900 lives. Two eruptions of this series injected $SO_2$ into the stratosphere, a requirement to get a long lasting stratospheric sulfate layer. The first eruption on March 17th injected 4.7 Tg $SO_2$ into the stratosphere, the second eruption 2.3 Tg $SO_2$ on May, 16th. In recent volcanic emission data sets these eruption phases are merged together to one large eruption

phase for Mt. Agung in March 1963 with an injection rate of 7 Tg $SO_2$. The injected sulfur forms a sulfate layer in the stratosphere. The evolution of sulfur is non-linear and depends on the injection rate and aerosol background conditions. We performed ensembles of two model experiments, one with a single and a second one with two eruptions. The two smaller eruptions result in a lower burden, smaller particles and 0.1 to 0.3 $Wm^{-2}$ (10 - 20%) lower radiative forcing in monthly mean global average compared to the individual eruption experiment. The differences are the consequence of slightly stronger

meridional transport due to different seasons of the eruptions, lower injection height of the second eruption and the resulting different aerosol evolution.

The differences between the two experiments are significant but smaller than the variance of the individual ensemble means. Overall, the evolution of the volcanic clouds is different in case of two eruptions than with a single eruption only. We conclude that there is no justification to use one eruption only and both climatic eruptions should be taken into account in future emission

datasets.

## 1   Introduction

In September 2017 Mt. Agung, a volcano on Bali, Indonesia (8.342°S, 115.58°E), became unrest. Earthquakes, steam, ash clouds and lahars resulted in the evacuation of nearly 150.000 people from the volcano's environment within a radius of 9-12 km in November 2017 (Gertisser et al., 2018). Finally, the eruption resulted in an ash cloud reaching up to an altitude of about

9.3 km (Marchese et al., 2018) and about 10 DU $SO_2$ above Bali (Hansen, 2017), which was not large and high enough to result in a climatic impact. The last climatic eruption of Mt. Agung dating back more than 50 years. From February 1963 to January 1964 a series of eruptions from Mt. Agung are documented (Fontijn et al., 2015). The initial unrest resulted in the 3rd largest eruption of the 20th century global volcano record and claimed about 1900 lives. Revising the literature, it is obvious that not only one of the eruptions was strong enough to inject $SO_2$ into the stratosphere, a requirement to get a long lasting stratospheric



sulfate layer, but also a second one (Self and Rampino, 2012). This second eruption injected 2.3 Tg $SO_2$ into the stratosphere on May, 16th, half of the 4.7 Tg $SO_2$ of the first eruption on March 17th (Self and King, 1996). The resulting sulfate layer caused a climatic impact by scattering and absorbing solar and terrestrial radiation leading to a temperature decrease of about 0.4 K in the tropical troposphere (Hansen et al., 1978).

Up to now, all recent volcanic forcing data sets assume one large eruption phase for Mt. Agung, the one in March 1963, but neglect the second one. Also the injection rate was merged to 7 Tg $SO_2$. Within this paper we will examine whether or not it is important to consider both eruption phases individually, when simulating sulfate evolution and transport, as well as the impact on radiative forcing of the Mt. Agung eruption.

The radiative forcing of the sulfate aerosols can either be simulated by calculating the evolution and transport of sulfur with

an aerosol microphysical model or, much simpler, by prescribing the optical parameters of the volcanic aerosols. The first needs volcanic injection data and information on emission strength and altitude, the latter optical properties like the aerosol optical depth (AOD). Most datasets base the estimated global coverage of sulfate aerosols after the Mt. Agung eruption on ground based and on ice core measurements, and provide the AOD (e.g. Sato et al. (1993), Stenchikov et al. (1998), Ammann et al. (2003), Crowley et al. (2008), Crowley and Unterman (2012)). Satellite data were yet not available in 1963. Newer datasets

rely not only on measurements, they also include simulated sulfate distributions, e.g. results of an empirical aerosol forcing generator like EVA (Toohey et al., 2016) or complex aerosol models, which simulate the evolution of the aerosol e.g. SAGE-4$\lambda$ (Arfeuille et al., 2014). On the other hand new volcanic eruption data sets are released which provide the $SO_2$ injection rate for large climate relevant volcanic eruptions, e.g. Volcan-EESM (Neely III and Schmidt, 2016) and the evolv2k data set (Toohey and Sigl, 2017) provide sulfur injection data. These newer datasets include one eruption phase only for Mt. Agung, the main

eruption in March 1963, and merged the mass of both eruptions.

The evolution of the volcanic aerosols is strongly non-linear. Especially the particle size depends on the erupted mass (Timmreck et al. (2010), Niemeier and Timmreck (2015)) and sulfur injected into an existing volcanic sulfate layer evolves differently than sulfur injected into background conditions (Laakso et al., 2016). Additionally, many chemical processes depend on particle size and sulfate concentrations. E.g. stratospheric OH, $NO_x$ and ozone concentrations change under high sulfur load.

CCMVal (2010) shows in Figure 8.20 the temperature response of different stratospheric chemistry models to volcanic sulfate aerosols. Models using full aerosol microphysics or prescribing surface aerosol density tend to overestimate the measured heating in the stratosphere after the Mt. Agung eruption. This might be related to the assumption of one eruption phase only, especially as for the Mt. Pinatubo eruption in 1991 the models show a slightly better result.

In this study we would like to address the following question: Is there a significant difference when simulating two medium

eruptions instead of a single large one only? We performed two experiments to provide an answer to this question. We describe the model and the simulations in more details in Section 2, show results in Section 3 where we describe the different burden results of the two ensembles (Sect. 3.1) and the cumulative impact of the eruptions (Sect. 3.2). Finally, we compare our results to measurements in Section 4 before we are concluding them in Section 5.



## 2 Model and observation data

### 2.1 Model setup

The model simulations of this study were performed with the middle atmosphere version of the general circulation model (GCM) MAECHAM5 (Giorgetta et al., 2006). The aerosol microphysical model HAM (Stier et al., 2005) is interactively

coupled to the GCM and was extended to a stratospheric version (Niemeier et al., 2009). MAECHAM5-HAM, ECHAM-HAM later in the text, was applied with the spectral truncation at wave-number 42 (T42), a grid size of about 2.8°, and 90 vertical layers up to 0.01 hPa. The model is not coupled to an ocean model and shows pure volcanic forcing response only.

HAM calculates the evolution of sulfate from the injected $SO_2$ to sulfate aerosol, including nucleation, accumulation, condensation, and coagulation and transport and sink processes like sedimentation and deposition (Stier et al., 2005). A simple

stratospheric sulfur chemistry is applied above the tropopause (Timmreck, 2001; Hommel et al., 2011) and the sulfate is radiatively active. The model setup is described in more details in Niemeier et al. (2009) and Niemeier and Schmidt (2017).

The L90 version of MAECHAM5-HAM generates interactively a quasi-biennial oscillation (QBO) (Giorgetta et al., 2006). However, we decided to nudge the QBO in the tropical stratosphere to the observed monthly mean winds at the Equator (updated Naujokat (1986)), as described in Giorgetta and Bengtsson (1999). This allows to inject the volcanic sulfur into

the observed QBO phase and still include the better resolved transport processes of the L90 version, e.g. a less permeable subtropical transport barrier (Niemeier and Schmidt, 2017). Nudging the QBO prescribes the feedbacks of the sulfate aerosol heating in the stratosphere on the QBO winds as observed. However, the QBO winds are prescribed on a monthly basis. This may suppress very short term changes in the transport due to dynamical changes caused by aerosol heating at the equatorial stratosphere.

### 2.2 Model simulations

We performed experiments of two scenarios for the 1963 eruption of Mt. Agung. We assumed for the first experiment one eruption phase at March 17th (AGUNG1) with an injection of 7 Tg $SO_2$ over three hours and for the second experiment two eruptions phases (AGUNG2): The first on March, 17th over three hours at an altitude of 50 hPa and a second on May, 16th over four hours at a slightly lower altitude of 70 hPa. The details of the eruptions were taken from Self and Rampino (2012) who

provided various information about both climatic eruptions of Mt Agung. The ECHAM5-HAM input data for the eruptions are summarized in Table 1.

We performed a set of six ensemble members for each eruption case. We initialized the model with two different states of the atmosphere and varied the stratospheric horizontal diffusion factor. This method is used regularly to disturb the atmosphere and create different states of the dynamical situation. All six simulations were used to calculate an ensemble mean.



**Table 1.** Overview over the performed simulations and information to the eruption details, after (Self and Rampino, 2012).

| Simulation name | Eruption mass | Eruption altitude | Eruption duration | Eruption date |
|---|---|---|---|---|
| AGUNG1 | 7 Tg $SO_2$ | 50 hPa | 6 - 9 UTC | March, 17th 1963 |
| AGUNG2 | 4.7 Tg $SO_2$ | 50 hPa | 6 - 9 UTC | March, 17th 1963 |
| | 2.3 Tg $SO_2$ | 70 hPa | 17 - 21 UTC | May, 16th 1963 |

## 2.3 Observations

We compare our simulation results to observations and volcanic data sets provided for the climate model intercomparison project (CMIP). The aerosol optical depth (AOD) prescribed for the years 1963 to 1964 in the CMIP5 simulations of the Max Planck Institute for Meteorology is based on Sato et al. (1993) and Stenchikov et al. (1998). The data rely mainly on astronomical observations summarized by Dyer and Hicks (1968), as no satellite data are available for the period. The AOD data for CMIP6 were taken from the SAGE-$3\lambda$ database (Luo (2016), Revell et al. (2017)). They combine ice core data (Gao et al., 2008) and AOD data (Stothers, 2001) with aerosol microphysical model simulations and include only one eruption of Mt. Agung in their preparatory model simulations (Arfeuille et al., 2014).

Stothers (2001) assembled a revised chronology of observed AOD after the Mt. Agung eruption. The data contain mainly measurements of atmospheric attenuation of starlight and direct sunlight. Stothers (2001) provides values of monthly mean AOD data as an average over different measurement sites, between 20°N to 40°N and 20°S to 40°S and results of monthly mean data of single measurement sites. Stothers (2001) excluded data of tropical measurement sites because they were not reliable.

Radiosonde temperature measurements provide information on the heating of the stratospheric aerosol layer after the eruption. This heating causes changes in stratospheric dynamics (Aquila et al. (2014),Toohey et al. (2013)) and is, therefore, an important value which should be taken into account correctly. Free and Lanzante (2009) provide vertical temperature anomalies after volcanic eruptions from radiosonde data (RATPAC). The carefully examined dataset contains data of 85 radiosonde stations, 32 of them in the tropics (Free et al., 2005). The QBO and ENSO signals in the temperature were removed. To calculate the temperature anomalies, the average of the two years before the eruption was subtracted from the average over the two years after the eruption. The corresponding model data of AGUNG1 and AGUNG2 were calculated by averaging over the two years after the first eruption and subtracting an ensemble mean of control simulations with nudged QBO data, but no volcanic eruption, for the period 1964 to 1965. This provides the anomaly and removes the QBO signal at the same time.

## 3 Results

Figure 1 shows the monthly and zonally averaged sulfate burden of the ensemble mean. Mt. Agung is located at 8° in the southern hemisphere (SH) tropics. Thus, the main transport direction of the aerosols is southward and, hence, burden values in





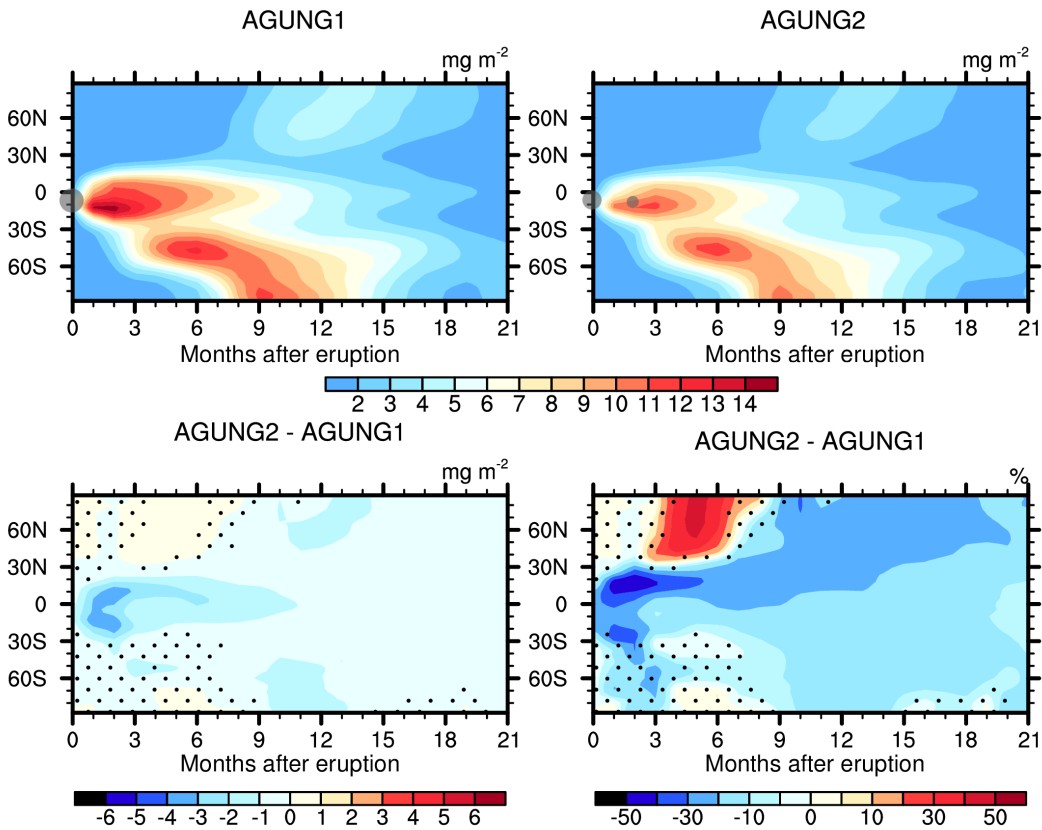

**Figure 1.** Top: Ensemble mean of sulfate burden of experiments AGUNG1 (left) and AGUNG2 (right). Bottom: Absolute (left) and relative differences (right) of the two ensembles. X-axis gives the months after the first eruption in March 1963. Stippling indicates non-significant differences at 99% level, following a student-t test. The gray dots mark the location and size of the volcanic eruptions.

the northern hemisphere (NH) remain small. The ensemble mean shows for both experiments, AGUNG1 and AGUNG2, two areas with high burden: a maximum in the southern tropics in the months 1 to 3 after the eruption, and about four months later in the SH mid- and high-latitudes with a secondary maximum between 30°S and 60°S. The maximum burden is slightly above 14 $mg/m^2$ in the ensemble of AGUNG1 and 10 $mg/m^2$, about 30% lower, in the ensemble of AGUNG2, reflecting the ratio of the initial injection. The initially higher injection in the ensemble of AGUNG1 results in a higher burden over almost all simulated months and regions. The strongest absolute difference between the two ensembles occurs in the time period between both climatic eruptions, when the injected sulfur amount in AGUNG2 is still smaller. The relative difference highlights that



more aerosols are transported into the NH tropics in AGUNG2 in the first months after the second eruption (months 4 to 6). The burden of AGUNG2 increases slightly after the second eruption but, overall, the tropical maximum of the burden is smaller and occurs later than in AGUNG1. In the SH extratropics the differences between the two ensembles are below 20%, 1 to 2 $mg/m^2$. Opposite, the burden is up to 50% larger in AGUNG2 in months 6 to 10 in the NH extratropics, poleward of 30°N,

but with small absolute values. Also towards NH winter the relative difference between the two simulations is larger in the NH than in the SH, which indicates differences in the transport regime and wind systems.

## 3.1 Transport of aerosols

The ensemble of AGUNG2 results in most areas and times in a lower sulfate burden than AGUNG1 (Fig. 1). An exception is the sulfate burden in the extratropics in the first months after the second eruption. This indicates a stronger meridional transport

in AGUNG2. The second eruption occurs two months later, thus, in a different season with different stratospheric transport pattern. Additionally, the injection height is lower, 70 hPa instead of 50 hPa. Figure 2 shows the monthly mean zonal wind (shaded) and the residual stream function for April and June 1963, one month after the eruption each. We show the results of the AGUNG2 simulation only, as the zonal wind and stream function is very similar between the two simulations. Nudging of the QBO at the Equator leads to only ± 10% difference of zonal mean zonal winds in the extratropics between the two

experiments.

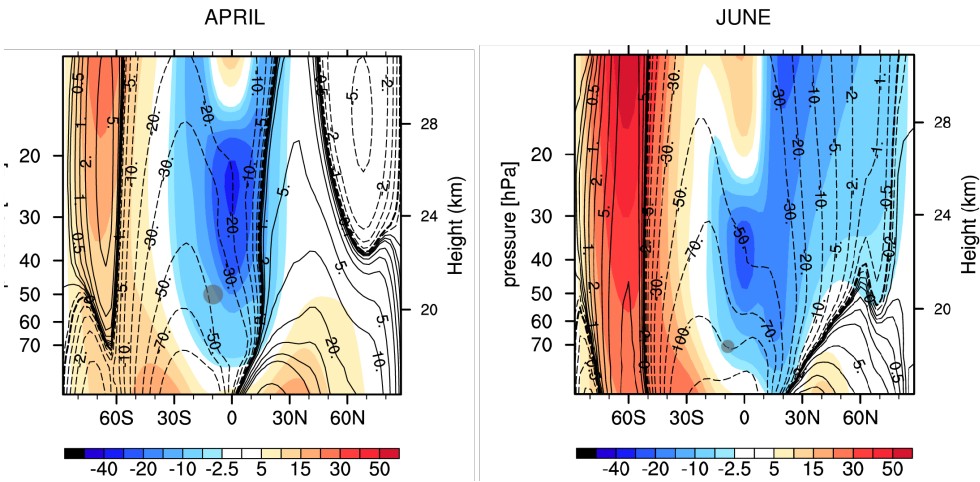

**Figure 2.** Monthly mean zonal wind [$ms^{-1}$] of AGUNG2 for April (left) and June (right) 1963 in shading. Contour lines show the stream function [$kg\,s^{-1}$]. Positive (solid) streamlines describe clockwise circulation, negative (dashed) ones counter-clockwise circulation. The gray dots mark the injection location of the two volcanic eruptions.

Punge et al. (2009) showed that meridional transport in the tropics and sub-tropics depends on the QBO phase. Figure 2 shows that both eruptions phases inject into easterly zonal wind. Thus, the QBO phase should not play an important role in transport characteristics of both experiments. Seasonality seems more important. The streamlines show that the zero-line,



indicating the tropical pipe, is shifted northward in June, allowing more sulfate to be transported into the NH. Additionally, the second eruption occurs at a lower altitude, 70 hPa, where the sub-tropical transport barrier is weaker than at 50 hPa. This allows more meridional transport than at 50 hPa: the stream line at 70 hPa in June is stronger than at 50 hPa in April, 100 $\mathrm{kgs^{-1}}$ and around 50 $\mathrm{kgs^{-1}}$, respectively.

## 3.2 Cumulative impact — a sum over time

The zonally averaged cumulative burden (Fig. 3, left), time integrated monthly mean values over 21 months, starting with the month of the first eruption, is roughly 20% lower for AGUNG2 than AGUNG1. The difference between the two results is larger in the tropics than in the secondary maximum around $50°\mathrm{S}$. This results from the stronger meridional transport towards the SH in AGUNG2. Thus, one eruption with larger $SO_2$ injection results in higher burden than the same injected amount of $SO_2$, but split into two eruptions and in a slightly stronger tropical confinement of the aerosols. However, the shaded areas indicate that the variance within each ensemble is larger than the differences between the ensemble mean values. This result is confirmed in

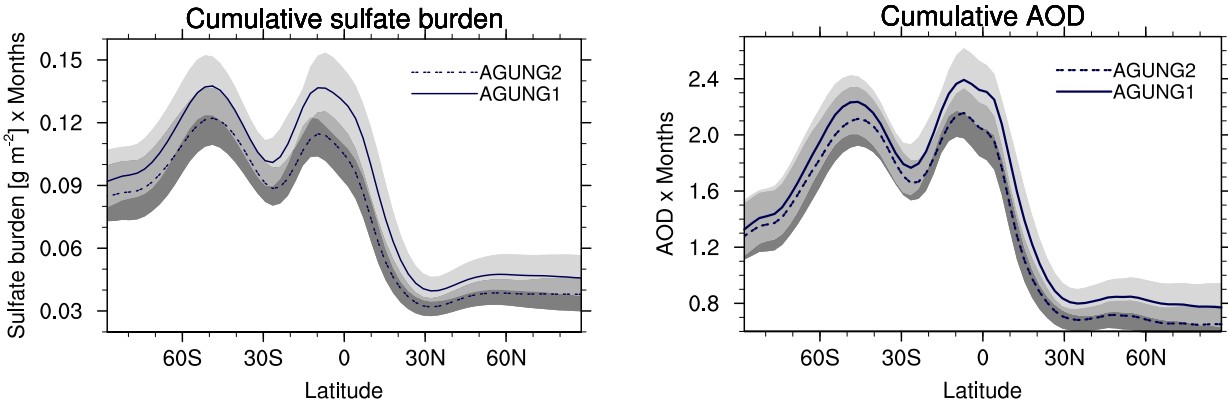

**Figure 3.** Cumulative values, integral over 21 months, of (left) zonally averaged sulfate burden and (right) AOD at 550 nm. The gray shadings indicate the maximum and minimum values of the single simulations in the ensemble, light gray for AGUNG1, dark gray for AGUNG2 and medium gray for the overlapping area.

the cumulative AOD (Fig. 3, right). But, the AOD of AGUNG1 and AGUNG2 differs less (about 10%) than the burden (about 20%) in the tropics and even less in the SH extra-tropics (about 6%). The reason for this are different particle radii. Scattering of sulfate aerosols decreases with increasing particle radius. The maximum effective radii of the sulfate aerosols reach 0.5 μm at $8°\mathrm{S}$ for AGUNG1 and stay below 0.45 μm for AGUNG2 (Figure 4, top). Sulfate particles are 0.05 μm smaller in AGUNG2. Thus, they scatter more intense and the AOD difference between the two experiments gets smaller. These smaller radii are the consequence of the lower injection rate of the first eruption in AGUNG2. Particle sizes increase with increasing injection rate (Niemeier and Timmreck, 2015). Laakso et al. (2016) simulated an volcanic eruption into background sulfate level and into elevated sulfate level from continuous injections for climate engineering (CE). They show a shorter lifetime and larger particles under CE conditions. Thus, following Laakso et al. (2016), we would expect stronger coagulation after the second eruption, as



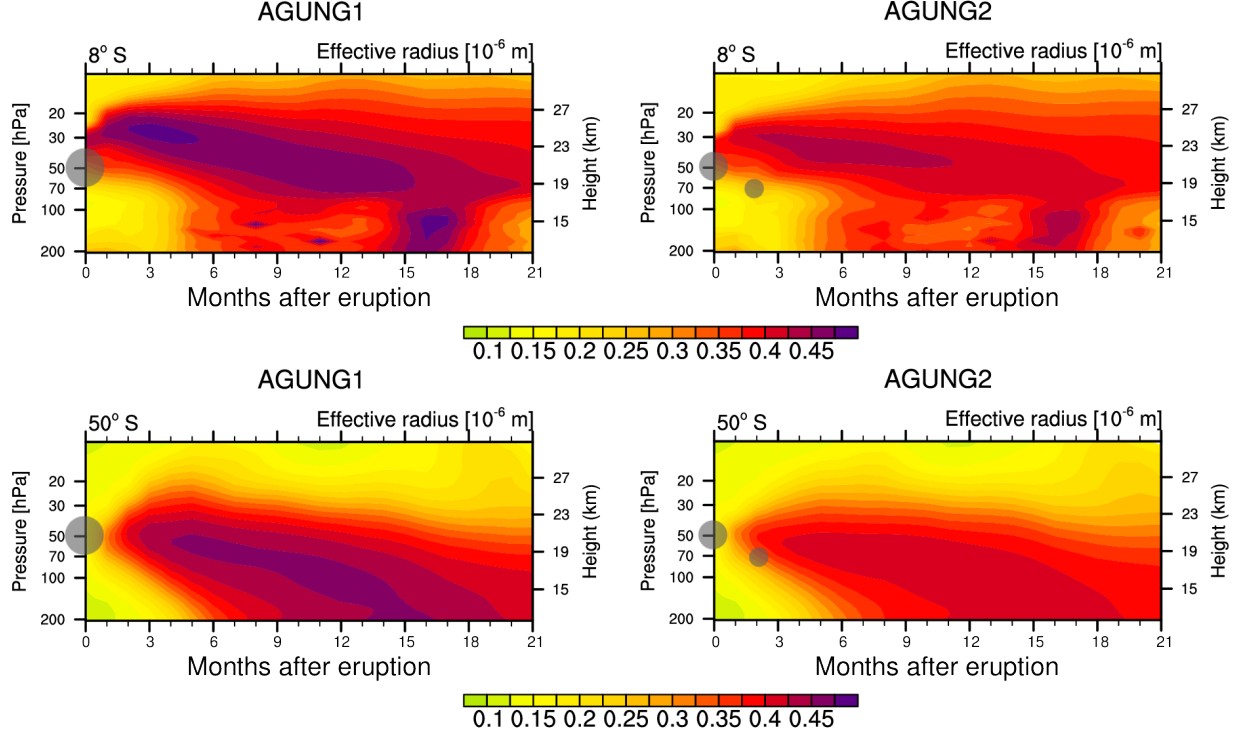

**Figure 4.** Hovmøller diagram of monthly mean effective radius μm of sulfate at the grid point corresponding to 8°S (top) and 50°S (bottom).

newly formed particles coagulate fast with available larger particles. But the second eruption occurs at lower altitude, where sulfate concentration and particle radii are smaller. Thus, coagulation is less important.

The climatic impact can be derived from the radiative forcing imbalance at top of the atmosphere (TOA). The global monthly mean TOA forcing of sulfate is about 0.1 to 0.3 $\mathrm{Wm}^{-2}$ larger in AGUNG1 (Figure 5, left). This difference is three to ten times larger than the radiative forcing of stratospheric ozone (-0.033 $\mathrm{Wm}^{-2}$) in CMIP6 (Checa-Garcia et al., 2018) and comparable to the radiative forcing of the total ozone column (0.28 $\mathrm{Wm}^{-2}$ in CMIP6). The forcing of anthropogenic sulfate aerosols is assumed as -0.4 $\mathrm{Wm}^{-2}$ (Stocker et al., 2013). Surface temperature changes $\Delta T_s$ relate to forcing as $\Delta T_s = \alpha F$, with $F$ the radiative forcing and $\alpha$ the climate sensitivity (Gregory and Webb, 2008; Ramaswamy et al., 2001). $\alpha$ is a constant which differs for each model. Thus in our case

$$\frac{\Delta T_s(AGUNG1)}{\Delta T_s(AGUNG2)} = \frac{F(AGUNG1)}{F(AGUNG2)} = \frac{-1.35\mathrm{Wm}^{-2}}{-1.23\mathrm{Wm}^{-2}} = 1.1, \tag{1}$$





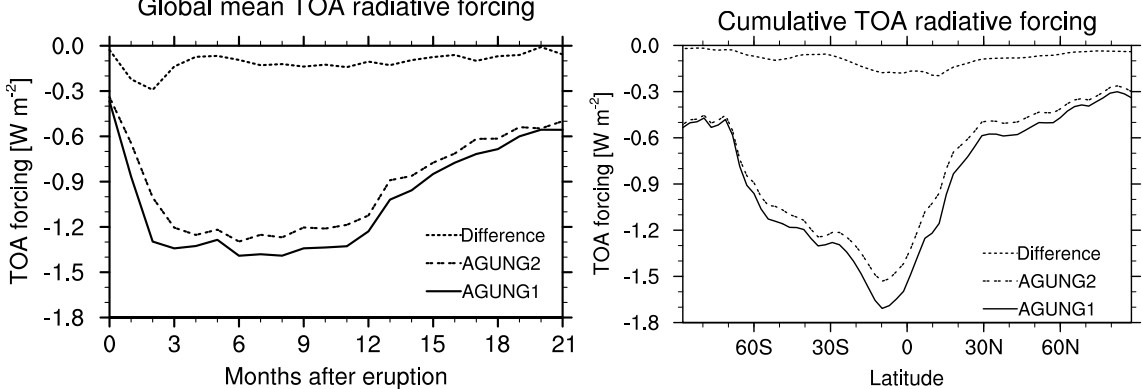

**Figure 5.** Top of the atmosphere (TOA) radiative forcing of sulfate aerosols under all sky conditions. Left: Global mean TOA forcing over time. Right: Zonally averaged radiative forcing as average over time (21 months)

.

with F(AGUNG1) and F(AGUNG2) the averages over the global radiative forcing of the months 3 to 9 after the eruption (Figure 5, left). Thus, we overestimate the cooling in AGUNG1 by a factor of 1.1 or 10%. Both experiments show the strongest difference of radiative forcing in the tropics (Figure 5, right) and the strongest surface cooling occurs in the tropics as well.

## 4    Comparison to observations

The zonally averaged AOD of AGUNG1 and AGUNG2 differs mainly in the tropics (Figure 6, top) and is rather similar in the SH extratropics. Our results agree quite well to the CMIP6 AOD, which shows, however, slower transport into the extra-tropics and no secondary maximum at 40°S to 50°S. Less aerosols reach the SH high-latitudes in CMIP6, but they have a longer lifetime. The CMIP5 volcanic forcing data used for the MPI-ESM simulations show a very different evolution which might be related to not reliable measurements in the tropics (Stothers, 2001).

Both AGUNG experiments fit in general well to the measurements of Stothers (2001), which are included as circles in Figure 6. The simulated NH AOD is slightly larger than the measurements. Thus, it seems that ECHAM-HAM overestimates the northward transport, but we have to take into account that Stothers (2001) noted the measured NH AOD is barely above noise level. In the SH the modeled AOD is smaller than the measurements, but larger than the CMIP6 data. In ECHAM-HAM, meridional exchange within the sub-tropics results in lower values between 20°S and 30°S and a maximum at 50°S where the

meridional transport is blocked by the edge of the polar vortex. This maximum AOD, above 0.2, is more similar to the measured AOD between 20°S and 40°S which could be an indication for too strong meridional transport in ECHAM-HAM. The AOD measurements of the months 19 to 21 and the CMIP6 data may indicate a too short lifetime of the simulated aerosols at the SH high latitudes. This is most probably related to too high wet deposition at high latitudes, a phenomena of ECHAM-HAM also



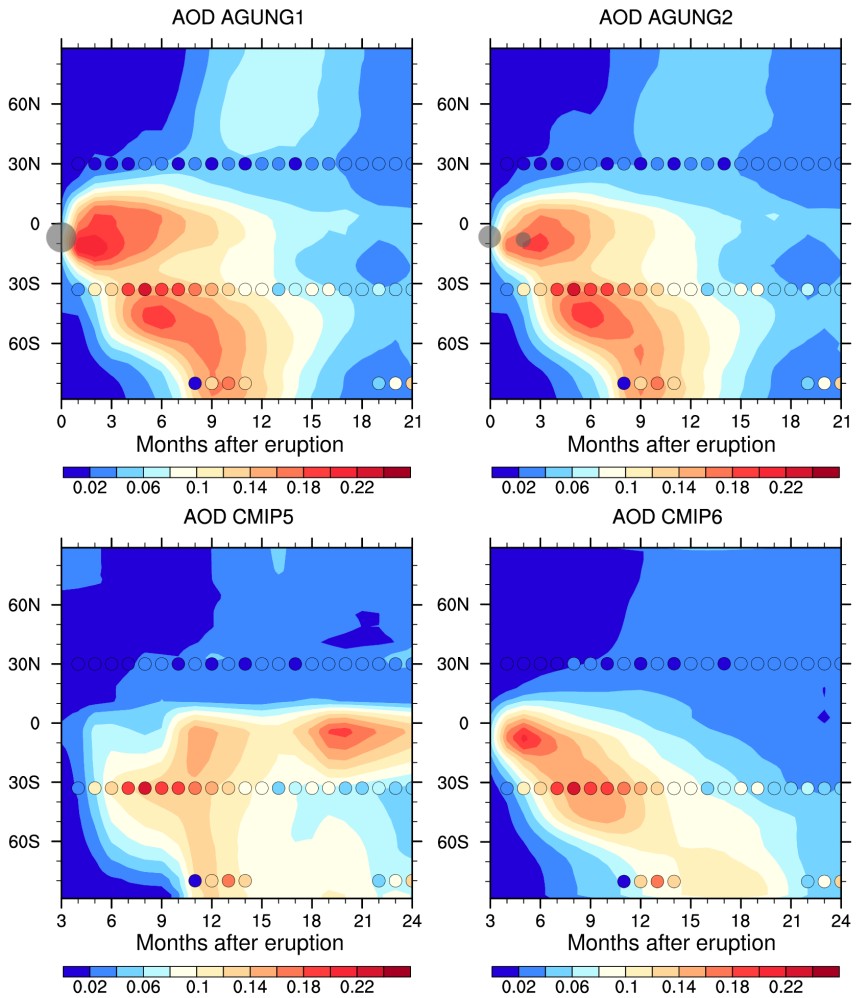

**Figure 6.** Monthly mean AOD at 550 μm over time. Top: AGUNG1 and AGUNG2 ensemble. Bottom: AOD used for CMIP5 simulations (Stenchikov et al., 1998) and AOD for CMIP6 simulations Luo (2016). Overlayed as colored circles are measurements of monthly mean AOD averaged over the regions 20°N to 40°N and 20°S to 40°S, given in Table 3 of Stothers (2001), and single values for the South pole (after Figure 2, same paper). The gray circles mark the volcanic eruptions, the size represents the size of the eruption.

reported by Toohey et al. (2013) and Marshall et al. (2018). Toohey et al. (2013) assume, again, too strong meridional transport in the stratosphere as a cause.

A more detailed analyses of the SH extra-tropics is shown in Figure 7. We averaged the model data over latitude bands 25°S to 30°S and 35°S to 40°S to compare those to the single measurement data given in Figure 1 of Stothers (2001). The simulated AOD is clearly smaller than the point measurements, but one should take into account that a point measurement gives a higher value than an area mean of the model. Additionally, measurements and simulated values depend not only on sulfate evolution but also on transport. Further possible reasons for the differences were stated before. In the first three months





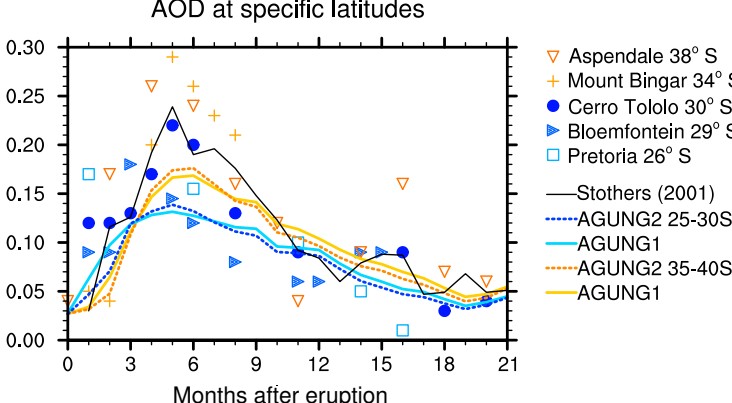

**Figure 7.** Monthly mean SH AOD (550 μm) over time. Colored lines show model results averaged over 25°S to 30°S (blue) and 35°S to 40°S (orange), each for both ensembles. The black line gives data of Stothers (2001), average of measurements between 20°S and 40°S. Single markers show single measurement data, estimated from Figure 1 in Stothers (2001), with a similar color code than the model data.

after the eruption, the simulated AOD agrees well to the measurements. The onset of the meridional transport is similar, but the simulated volcanic cloud arrives slightly later in 25°S to 30°S (see also Fig 6). Between 35°S to 40°S the model agrees better to the data of Mt. Bingar (yellow cross) than to the Aspendale data (red triangle) in the first months after the eruption, where the sulfate values increase two months earlier. Both, the timing of the maximum and the onset of the decline agrees well

in measurements and model results.

Measured data of particle radii are not available, but the following references provide some estimates, which are helpful for a rough comparison. Arfeuille et al. (2014) simulated for the SAGE-4λ dataset effective radii of 0.3 μm to 0.4 μm for eruptions of the size of the the Agung eruption. Stothers (2001) estimates a radius of 0.35 μm from the measurement data of the year 1963 at Bloemfontain, South Africa (29°S). Our simulated radii at the Equator (above 0.45 μm) and at 50°S (above 0.4 μm) are larger

than these values for both experiments (Figure 4), with a better agreement in AGUNG2. A smaller radius of the aerosols would cause a larger AOD. The larger radii in our experiments lead us to the question of the requirement to include an OH-limitation process for modeling the sulfate evolution, which is a still open research question. In case of high $SO_2$ concentrations OH might be limited for further $SO_2$ oxidation(Bekki et al., 1996). The slower formation of sulfuric acid vapor would result in smaller particles. Bekki et al. (1996) assumed that OH limitation occurs only in case of a super-eruption, but Mills et al. (2017)

show OH-limitation also after the Mt. Pinatubo eruption. Contradictory, LeGrande et al. (2016) showed that water vapor in the eruption cloud increases the amount of available OH. This reaction was not included in the two studies cited above. The here presented results use a fixed monthly mean OH concentration which is not influenced by the volcanic cloud. Following Mills et al. (2017), this missing OH limitation leads to faster formation of sulfate resulting in larger sulfate particles. Therefore, we performed one simulation with a simple parameterization of OH limitation, see the supplementary materials for details. This

simulation shows slower sulfate formation, which agrees less to the measurements, and only a slightly higher AOD half a year





after the eruption (Figure SM1 and SM2) for the OH-limited case. As we use a simplistic parameterization these results are very limited.

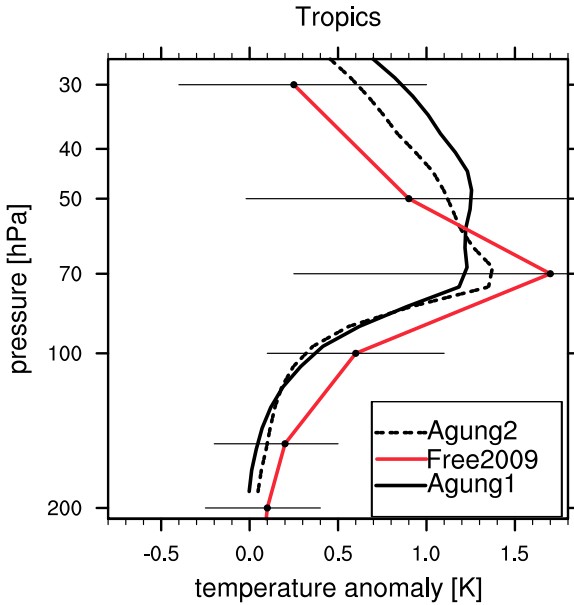

**Figure 8.** Profile of temperature anomaly [K] compared to RATPAC radiosonde measurements taken from (Free and Lanzante, 2009) in the tropics (30°N to 30°S). Model results and measurements are averaged over two years after the eruption.

Sulfate aerosol absorbs terrestrial radiation and warms the stratosphere. This temperature signal depends in the model on the coupling to radiation. We compare the results of the ensemble mean temperature data to radiosonde data of Free and Lanzante
(2009). Both, model and measurement data are independent of QBO temperature relations (see Section 2.3). The simulated temperature anomaly is 0.2 K to 0.3 K smaller below 70 hPa and up to 0.5 K larger above 50 hPa (Figure 8). Thus, the simulated sulfate layer is lifted to higher altitudes than the measurements indicate. Additionally, the ozone concentration in ECHAM-HAM is not impacted by the volcanic sulfate. This missing ozone response on high sulfate concentrations may cause a temperature offset in the ozone layer. AGUNG2 shows a clear maximum at 70 hPa, similar to the radiosonde data, and a
better vertical distribution of the temperature anomaly. While theses results compare well to the radiosonde data, simulated temperature anomalies are only half of the observed one in the SH (not shown). This may hint towards a too short lifetime of the simulated sulfate aerosols (see Figure 6 too).

## 5 Summary and conclusions

We compared results of two scenarios for the Mt. Agung eruption in 1963: the commonly used one eruption scenario with a
15 strength of 7 Tg SO$_2$ and the observed scenario with two climatic eruptions of 4.7 Tg SO$_2$ and 2.3 Tg SO$_2$ respectively. We





simulated lower burden in the tropics, but slightly stronger meridional flow into the extratropics in the simulation with two eruptions, AGUNG2. We relate the stronger meridional flow to the lower injection height of the second eruption, where the tropical transport barrier is weaker. Additionally, the position of the tropical pipe is further northward in May than in March. This allows more aerosols to be transported into the NH. The smaller injection rate and the two different injection heights cause the particle to grow less than in AGUNG1. These processes result in 10% to 20% lower radiative forcing, or 0.1 to 0.3 $\mathrm{Wm}^{-2}$ in monthly mean global average, and estimated 10% less surface cooling in AGUNG2. The strongest signal would occur in the tropics. However, the difference in the climate impact between the two experiments is smaller than the spread of the single ensemble members.

When comparing to the few available measurements we see that the differences to the measurements are larger than the differences between the two experiments. We seem to underestimate the observed AOD and simulate larger particle radii. The timing of the aerosol evolution in the model seems to be supported by the measurements. Given the low number of observations at that time, especially in the tropics, it is difficult to validate the two experiments. Overall, the smaller particle size and better located maximum temperature anomaly in the vertical profile of AGUNG2 are consequences of different transport and microphysical processes between the two experiments. These are arguments to include both climatic eruptions into future emission datasets.

One could also argue that the large model spread, as e.g. described by Marshall et al. (2018) and Zanchettin et al. (2016), limits the interpretation of our model results. Other models may get quantitatively different results, but most probably the simulated difference between the two eruption scenarios would be qualitatively similar: A lower radiative forcing of the Agung 1963 eruption when including two eruption phases. Including a more sophisticated atmospheric chemistry (including OH chemistry, water vapor and ozone) may increase the differences between the scenarios. Lower $SO_2$ injection rates in AGUNG2 would cause less impact on OH and ozone, thus on chemical species in the stratosphere. Taking two eruptions phases into account will be important for processes in the early evolution of sulfate. Ash and ice are important species in this early phase. Both were not taken into account in the here presented simulations but are planed for the future.

Overall, differences of around 10% in the radiative forcing between AGUNG1 and AGUNG2 could justify taking only a single eruption phase of the 1963 Mt. Agung eruption into account. On the other hand, the more recent volcano datasets are rather detailed. Also future studies using high horizontal and vertical resolution and more sophisticated models will demand detailed input data. Therefore, a single eruption should not be justified and we recommend to include both eruptions of Mt. Agung in upcoming datasets.

*Acknowledgements.* We thank Hauke Schmidt for implementing the QBO nudging into ECHAM5-HAM and Elisa Manzini for valuable comments. A discussion about the OH-limitation in WACCM with Simone Tilmes helped to modify ECHAM5-HAM. This work is a contribution to the the European Union project StratoClim (FP7-ENV.2013.6.1-2) and the German DFG-funded Priority Program 'Climate Engineering: Risks, Challenges, Opportunities?' (SPP 1689). U. Niemeier is supported by the SPP 1689 within the project CELARIT and by StratoClim . Claudia Timmreck acknowledge support from the German federal Ministry of Education (BMBF), the research program



MiKlip (FKZ:01LP1517(CT):/01LP1130B(MT)) and from the European Union project StratoClim. The simulations were performed on the computer of the Deutsches Klima Rechenzentrum (DKRZ). Primary data and scripts used in the analysis and other supplementary information that may be useful in reproducing the author's work are archived by the Max Planck Institute for Meteorology and can be obtained by contacting publications@mpimet.mpg.de. Model results will be available under cera-www.dkrz.de soon.



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
