# Peer review of "Revisiting the Agung 1963 volcanic forcing — impact of one or two eruptions"

_Atmospheric Chemistry and Physics, 2019_

## Referee Comment (RC1) · Alan Robock (Referee) · 28 May 2019

The paper presents interesting results, and I recommend it be published after the authors address the points below.

The results in this paper seem correct, but there are several issues discussed in the 50 comments in the attached annotated manuscript that should be addressed. The most important ones are:

1. The 2-eruption scenario changed both the amount and altitude of the sulfur injections. Thus, it is difficult to separate those effects. If you change two things at once, it is harder to understand the causes of the differences. Why not just change the number of eruptions, without changing the altitude, also?

[Figure]

2. I don't understand how the ensembles were created. The authors said there were two different initial conditions. They did 6 ensemble members. Why did they use 2 different atmospheric states? And what were the states? Did they use actual weather patterns observed at the time of the eruption, from a reanalysis? If not, how they you choose them? And why did they only do 6 ensemble members?

3. What SSTs were used? Observations?

4. I would like to see the climate responses compared to each other, and to observations. What about surface air temperature patterns? Precipitation patterns? Northern and Southern Annular Modes? Did the injection strategies produce significant differences?

5. Except for Fig. 3, I did not see any statistics showing the spread from the ensemble members. Rather the authors argue at the end that the differences between different models are larger than then differences shown here between the two forcings. But how do those differences compare to chaotic differences in the climate system? How do weather differences, or differences in initial conditions, or different oceanic responses compare to the differences from different forcings?

Review by Alan Robock

Please also note the supplement to this comment:
https://www.atmos-chem-phys-discuss.net/acp-2019-415/acp-2019-415-RC1-supplement.pdf

---

## Referee Comment (RC2) · Anonymous Referee #2 · 3 Jun 2019

Overview: The manuscript by Nemeier et al. aims at clarification of the stratospheric aerosol forcing after Mt. Agung eruption in 1963. Authors use a climate model with interactive aerosol microphysics and simplified stratospheric chemistry and look at the differences between runs specifying the eruption as a single injection and as two injections, with the latter being more precise but mostly not considered in previous studies. The results of ensemble simulations are compared to each other and to the available limited observations. Authors demonstrate that the two-injection scenario provides about 10% lower signal in different aerosol parameters and is in a better agreement with temperature observations.

The manuscript is well written, methods are clear and sufficient for the paper goals. Even though the model (as any other model) has some limitations, they are fairly dis-

cussed and don't reflect on the main conclusions. The results are important for clarification of the historical volcanic forcing and also present an interesting example of the stratospheric aerosol layer behavior for future studies. I suggest to publish this paper after addressing few minor comments.

Comments:

P3L7: Not clear which ocean you used - was it a climatology or historical variability.

P6L14: 10% difference in the stratospheric vortex zonal wind speed is not so small to be so easily discarded, given that you already noted the traces of stronger meridional transport. Was it statistically significant? Was the change in stream function also around 10% or negligible?

P8L4-7: It is a bit incorrect to compare rather short-term effects after eruption to the long-term forcings from ozone and anthropogenic aerosols without specifying this difference.

P9L18: Too high wet deposition or gravitational sedimentation? Wet deposition implies precipitation, which occurs in the troposphere and is already quite fast compared to the stratospheric aerosol lifetime. In your case it looks more like a quicker sedimentation from the stratosphere to the troposphere.

P11L2-4: Mount Bingar data agrees better with the model than the Aspendale data only in the first months, while later (months 6,8,10) it is the other way around.

P13L24-28: First you say that 10% justifies no importance and then the opposite. I would suggest rephrasing it in a simpler non-contradictive way, because your main message is still that it is important and actually does not require a lot of efforts to apply.
* * *

---

## Author Comment (AC1) · 17 Jul 2019

**Answers to reviewers on the ACPD paper (acp-2019-415):**
**Revisiting the Agung 1963 volcanic forcing — impact of one or two eruptions**

Ulrike Niemeier, Claudia Timmreck and Kirstin Krüger

Max Planck Institute for Meteorology, Bundesstr. 53, 20146 Hamburg, Germany

We thank Alan Robock and an anonymous reviewer for their helpful comments. We considered the recommendations carefully and made some changes in the text. The questions are in bold, answers in black and changes in the text in blue.

We realized an error in Fig. 8, the curves were averaged over different areas. This is corrected and a second set of curves, avereaged over the inner tropics, plus a discussion is added. The text changes to: The measurements, averaged over stations between 30°;N and 30°S, show a strong maximum at 70 hPa (Figure 8). The average over 30°N to 30 °S of the model results (black lines) show a smaller anomaly and a stronger vertical extension of the heated area, more in AGUNG1 than in AGUNG2. Below the maximum at 70 hPa the simulated temperature than the measurements. Both features indicate too strong vertical lofting in the model. Figures 6 and 7 indicated a too low AOD in the model results around 30°S. Therefore, we added a second temperature profile, averaged over the main volcanic cloud at 15°N to 15°S (gray line), to Figure 8. Now the maximum is represented better. More important is the better agreement with the temperature decrease between 50 hPa and 70 hPa, especially for AGUNG2. The easterly phase of the QBO is related to downward motion and suppresses the vertical lofting caused by the heating, but is also also related to stronger vertical transport in the secondary meridional circulation around 30°North and South. This up welling seems to be stronger in the model than in the measurements causing the too high vertical extension of the simulated cloud.

**Answers to reviewer1**

We thank Alan Robock for his inspiring comments. We considered the recommendations carefully and performed an additional experiment: We changed the AGUNG2 setup and injected $SO_2$ twice at 50 hPa in a single simulation. The results are shortly discussed in the supplementary material. Further, we followed the recommendations on grammar etc. of the commented pdf-file and included some remarcs in the following list of answers.

**Did they (Self and King 1996) actually apportion the sulfur injections like this? The abstract only gives the total of 7 Tg.**

Self and King (1996) give a total estimate of 7 to 7.5 Tg $SO_2$ as injection rate into the stratosphere for the Agung eruption, which they derived from stratospheric optical depths measurement a couple of months after the eruptions. The two eruptions in March and May were too close to estimate the SO2 emission of the observed two climatic eruptions with this method and satellite data were not available as nowadays.

In their table 3 Self and Rampino (2012) give a summary of the observed mass eruption rates, the length of the eruption and the estimated maximum column height of the eruption cloud. They obtain

values of $4.10^7$ kg s$^{-1}$ and $2.10^7$ kg s$^{-1}$ for the first and second eruption, and a quite similar eruption length. Hence we assume for simplicity that the second eruption was half of the size of the first one. This could only be an assumption as no injection rate for sulfur was given.

For clarification we also revised the text at several places , see below:

Abstract: The estimated mass flux of the first eruption was about twice as large as the mass flux of the second eruption. We followed the estimated emission profiles and assumed for the first eruption on March 17th an injection rate of 4.7 Tg SO$_2$, and 2.3 Tg SO$_2$ for the second eruption on May 16th.

Introduction: The mass flux of the second eruption on March 16th was about half the size of the mass flux of the first eruption on March 17th. Self and Rampino (2012) estimate the volumetric eruption rate for March 17th to be $\sim 1.8 \times 10^4$ m$^3$s$^{-1}$ over $\sim$ 3.5 h duration. The volumetric eruption rate of the May 16th event was $\sim 0.9 \times 10^4$ m$^3$s$^{-1}$ over $\sim$ 4 h duration. The resulting sulfate layer caused a climatic impact by scattering and absorbing solar and terrestrial radiation leading to a temperature decrease of about 0.4 K in the tropical troposphere (Hansen et al., 1978).

The sulfate load of the Mt. Agung eruptions was estimated to 7 to 7.5 Tg SO$_2$ from observations of aerosol optical depth (AOD) a couple of months after the eruption (Self and King, 1996). It was impossible to distinguish between single eruptions with this method. This could be the reason, that up to now, recent volcanic forcing data sets assume one large eruption phase of 7 Tg SO$_2$ for Mt. Agung, the one in March 1963, but neglect the second one.

Simulations: We performed experiments of two scenarios for the 1963 eruption of Mt. Agung. We assumed for the first experiment one eruption phase at March 17th (AGUNG1) with an injection of 7 Tg SO$_2$ over three hours. For the second scenario two eruptions were simulated with an ratio of the injection rate of 2:1. This reflects the mass flux of 4 kg s$^{-1}$ and 2 kg s$^{-1}$ given in Table 3 in Self and Rampino (2012). The heights of the eruptions were taken as average of the range of estimate heights in Self and Rampino (2012). This resulted for an assumption for the second experiment with two eruptions phases (AGUNG2): The first on March, 17th over three hours and a injection rate of 4.7 Tg SO$_2$ at an altitude of 50 hPa and a second on May, 16th over four hours and a injection rate of 2.3 Tg SO$_2$ at a slightly lower altitude of 70 hPa.

**The 2-eruption scenario changed both the amount and altitude of the sulfur injections. Thus, it is difficult to separate those effects. If you change two things at once, it is harder to understand the causes of the differences. Why not just change the number of eruptions, without changing the altitude, also?**

The injection height was estimated to 50 hPa for the first eruption and 70 hPa for second eruption, following the values of column heights of 18 to 23 km and 16 to 21 km (Self and Rampino (2012), Table 3).

Your comment has inspired us to do an additional sensitivity experiment: We changed the AGUNG2 setup and injected SO2 also at 50 hPa. We briefly discuss this in section 3.2 and added the results to the supplementary material.

We added in section 3.2: We performed an additional sensitivity study where we injected the SO$_2$

in both eruptions at 40 hPa to differentiate better between injection rates and emission height. The simulation AGUNG2 50hPa, with both eruptions at the same altitude, results in particle radii very similar to AGUNG1 and about 0.05 μm larger than in AGUNG2. This reflects the results of Laakso et al. (2016). See Section 2 in supplementary material for more information and figures.

**I don't understand how the ensembles were created. The authors said there were two different initial conditions. They did 6 ensemble members. Why did they use 2 different atmospheric states? And what were the states? Did they use actual weather patterns observed at the time of the eruption, from a reanalysis? If not, how they you choose them? And why did they only do 6 ensemble members?**

We have not nudged meteorological parameters to the real meteorological situation because the model should respond to the different radiative forcing scenarios. Hence, we nudged the zonal winds in the tropical stratosphere, the QBO.

We found two situations with a QBO phase comparable to the observation in the control run. We used these as start for two members of our ensemble. We further enlarged the ensemble by slightly varying the vertical diffusion parameter of to four different values in one of the original members. This resulted in four additional different meteorological situations, so 6 ensemble members in total.

We assumed this number to give a certain range of results. We prescribe fixed SST values and, thus, no variations in the ocean. Therefore, we expect our small ensemble to describe the variations in the limited range of solutions and would not expect a substantially different result from a larger ensemble.

We added a section to the supplementary materials: We performed a set of six ensemble-members for both, AGUNG1 and AGUNG2. We initialized the model from two different years, taken from a control simulation which was performed with ECHAM5-HAM under background conditions. Both years show a QBO phase similar, but still different, to observations before the Agung eruption. We started all simulations of the ensemble in 1962 with nudged QBO. We enlarged the ensemble by setting in 1962 the factor by which stratospheric horizontal diffusion is increased from one level to next level above to 1.001, 1.0001, 0.999, 0.9999, respectively, instead of 1. This method is used regularly to disturb the atmosphere and creates a different state of the dynamical situation. Thus, we got six different states of the atmosphere to start our volcanic eruption simulations. Finally, all six simulations were used to calculate an ensemble mean. The sulfate burden simulated in the single simulations of the ensemble are given in Figure 1.

We shortened the text in the main paper to: We performed a set of six ensemble members for each eruption case. See supplementary material for further details.

**What SSTs were used? Observations?**

We have used climatological SSTs, as we did in many previous studies for volcanic eruptions and geoengineering. Observed SSTs would have reflected the observed situation better. On the other hand is the model with climatological SSTs independent of any real situation — observational SSTs reflect the situation with two eruptions.

We added to the model description:

The sea surface temperatures (SST) are set to monthly mean climatological values based on the Atmospheric Model Intercomparison Project (AMIP) SST observational data set (Hurrell et al., 2008). Thus, the SST does not reflect the historical date but rather represents a climatological mean.

**I would like to see the climate responses compared to each other, and to observations. What about surface air temperature patterns? Precipitation patterns? Northern and Southern Annular Modes? Did the injection strategies produce significant differences?**

It is difficult to discuss the requested variables with a GCM. We discuss the climatic impact briefly when we estimate the impact of the different forcings on the surface temperature. The prescribed SSTs, especially climatological ones, suppress a realistic simulation of surface climate change, e.g. precipitation. This could be done with an Earth-system model but would be out of the scope of this study. Therefore, we decided to keep the current discussion on stratospheric impacts only.

**Except for Fig. 3, I did not see any statistics showing the spread from the ensemble members. Rather the authors argue at the end that the differences between different models are larger than then differences shown here between the two forcings. But how do those differences compare to chaotic differences in the climate system? How do weather differences, or differences in initial conditions, or different oceanic responses compare to the differences from different forcings?**

Figure 3 shows the spread of the ensemble minimum and maximum values and indicates that the spread is larger than the difference in the ensemble mean. We used this figure for the argument in the conclusions. This was not fully correct as we discuss the forcing in the conclusion. We added a $2\sigma$ variability range in Figure 5 (forcing) and deleted the sentence at the end. Additionally, a reference to Fig. 1 of the supplementary material was added, which shows the single burden results. This reference was missed in the main text.

We tried to include the chaotic system when performing an ensemble (see answer to point 2). A GCM with a fixed SST cannot include different states of the ocean, beside prescribing different SSTs, which would result in a different topic of the article.

**In 1963 there were no CFCs in the stratosphere. What would this response (on ozone) have been?**

We decided to skip the sentence on the missing ozone impact in ECHAM-HAM in the text. Checking the literature again, we decided that a full discussion of ozone changes after a volcanic eruption would be out of the scope of this paper.

CFC emission started prior to 1963. But in 1963 CFCs concentration was still small. However, clorine impacts ozone at low temperatures and is more important at higher latitudes than in the tropics. In the tropics, changes in the NOx/NOy equilibrium impact ozone more strongly than clorine (Richter et al., 2017).

Tie and Brasseur (1995) show in their Figure 1 very small changes in the column ozone abundance in the tropics in response to a Mt. Pinatubo-like volcanic eruption, calculated for different chlorine

loadings. However, there is also recent increasing observational evidence that volcanic halogens can reach the upper troposphere and lower stratosphere. Model simulations show that this can cause long-lasting impact on the ozone layer (Brenna Hans et al., 2019) with decreases above 10% in the tropical troposphere.

**What are the horizontal lines at each pressure level in Fig. 8?**

Thanks for mentioning, we forgot to describe this in the capture.

We added to the capture: The horizontal lines mark the 95% confidence interval for the RATPAC observations.

**Answers to reviewer 2**

We thank the reviewer for the helpful comments. We considered the recommendations carefully and made some changes in the text.

**P3L7: Not clear which ocean you used - was it a climatology or historical variability** Sorry, we should have mentioned this. We have used climatological SSTs, as we did in many previous studies for volcanic eruptions and geoengineering. Observed SSTs would have reflected the observed situation better. On the other hand is the model with climatological SSTs independent of any real situation. We added in the text:

The sea surface temperatures (SST) are set to monthly mean climatological values based on the Atmospheric Model Intercomparison Project (AMIP) SST observational data set (Hurrell et al., 2008). Thus, the SST does not reflect the historical date but rather represents a climatological mean.

**P6L14: 10% difference in the stratospheric vortex zonal wind speed is not so small to be so easily discarded, given that you already noted the traces of stronger meridional transport. Was it statistically significant? Was the change in stream function also around 10% or negligible?**

The difference in the zonal wind is not statistically significant. Differences in the stream function are around 5 kg s$^{-1}$ in the SH polar vortex. We added the figure for AGUNG1 to the supplementary materials.

We changed the text to: Figure 2 shows the monthly mean zonal wind (shaded) and the residual stream function for April and June 1963, one month after the eruption each, of AGUNG2 (see Fig. 5 in supplementary material for results of AGUNG1). Nudging of the QBO at the Equator results in similar zonal mean zonal winds in the tropics between the two experiments. Differences in the extratropics of about $\pm$ 10% are not significant and are mainly caused by a meridional shift of the higher latitude wind systems.

**P8L4-7: It is a bit incorrect to compare rather short-term effects after eruption to the long-term forcings from ozone and anthropogenic aerosols without specifying this difference.**

The reviewer is right with this comment. We shortened the text and include the discussion on temperature impact only. We changed the text accordingly:

The climatic impact can be derived from the aerosol radiative forcing at top of the atmosphere (TOA),

which was calculated with a radiation double call (Figure 5, left). The spread of the single ensemble members is large, but the average of the AGUNG2 ensemble is just out of the $2\sigma$ ensemble variability of AGUNG1. The global monthly mean TOA forcing of sulfate is about 0.1 to 0.3 Wm$^{-2}$ larger in AGUNG1. The average difference in the short-term volcanic forcing over the 1st 21 post eruption months is three to ten times larger than the long-term forcing radiative forcing of stratospheric ozone (-0.033 Wm$^{-2}$ ) in CMIP6 (Checa-Garcia et al., 2018) and comparable to the long-term radiative forcing of the total ozone column (0.28 Wm$^{-2}$ in CMIP6). The long-term radiative forcing of anthropogenic sulfate aerosols is assumed as -0.4 Wm$^{-2}$ (Stocker et al., 2013).

**P9L18: Too high wet deposition or gravitational sedimentation? Wet deposition implies precipitation, which occurs in the troposphere and is already quite fast compared to the stratospheric aerosol lifetime. In your case it looks more like a quicker sedimentation from the stratosphere to the troposphere.**

We agree with the reviewer. However, the cited papers compare deposition and show too high values for our model at high latitudes. The reason is more complicated and included sedimentation and precipitation. Sedimentation has to be strong to get the particles into the troposphere. Partly, the reason could be the model resolution (T42). Brühl et al. (2018) show a longer lifetime of sulfate aerosols after volcanic eruptions in T63 resolution, compared to T42, due to better representation of convection.

We changed the text to:

This is most probably related to too intense sedimentation at high latitudes (Brühl et al., 2018). The consequence is too high wet deposition, a well known phenomena phenomena of ECHAM-HAM.

**P11L2-4: Mount Bingar data agrees better with the model than the Aspendale data only in the first months, while later (months 6,8,10) it is the other way around.**

Yes, we agree. Unfortunately the measurements at Aspendale end 8 months after the eruption. The measurements at Mt. Bingar show quite large variations later.

We highlight the uncertainty related to the early measurements in the text:

The agreement between the model simulations and the individual stations differ with time. Between 35°S to 40°S, the model agrees better with the data of Mt. Bingar (yellow cross) than to the Aspendale data (red triangle) in the first months after the eruption, where the sulfate values increase two months earlier although the station is closer to the tropics. This may indicate the dependency of the point source to the position of the volcanic cloud.

**P13L24-28: First you say that 10% justifies no importance and then the opposite. I would suggest rephrasing it in a simpler non-contradictive way, because your main message is still that it is important and actually does not require a lot of efforts to apply.**

We changed the last paragraph to a more clear statement:

Overall, differences of around 10% in the radiative forcing between AGUNG1 and AGUNG2 should justify changes in the volcanic emission datasets. The more recent volcano datasets are rather detailed.

Also future studies using high horizontal and vertical resolution and more sophisticated models will demand detailed input data. Details of our assumptions on the Agung eruptions might be critically reviewed again, but we recommend to include both eruptions of Mt. Agung in upcoming datasets.

**References**

Brenna Hans, Kutterolf Steffen, and Krüger Kirstin: Global ozone depletion and increase of UV radiation caused by pre-industrial tropical volcanic eruptions, Scientific Reports, 9, 9435, https://doi.org/ https://doi.org/10.1038/s41598-019-45630-0, 2019.

Brühl, C., Schallock, J., Klingmüller, K., Robert, C., Bingen, C., Clarisse, L., Heckel, A., North, P., and Rieger, L.: Stratospheric aerosol radiative forcing simulated by the chemistry climate model EMAC using Aerosol CCI satellite data, Atmospheric Chemistry and Physics, 18, 12 845–12 857, https://doi.org/10.5194/acp-18-12845-2018, 2018.

Checa-Garcia, R., Hegglin, M. I., Kinnison, D., Plummer, D. A., and Shine, K. P.: Historical Tropospheric and Stratospheric Ozone Radiative Forcing Using the CMIP6 Database, Geophysical Research Letters, 45, 3264–3273, https://doi.org/10.1002/2017GL076770, URL https://agupubs.onlinelibrary.wiley.com/doi/abs/10.1002/2017GL076770, 2018.

Hansen, J. E., Wang, W.-C., and Lacis, A. A.: Mount Agung Eruption Provides Test of a Global Climatic Perturbation, Science, 199, 1065–1068, https://doi.org/10.1126/science.199.4333.1065, URL http://science.sciencemag.org/content/199/4333/1065, 1978.

Hurrell, J. W., Hack, J. J., Shea, D., Caron, J. M., and Rosinski, J.: A New Sea Surface Temperature and Sea Ice Boundary Dataset for the Community Atmosphere Model, Journal of Climate, 21, 5145–5153, https://doi.org/10.1175/2008JCLI2292.1, URL https://doi.org/10.1175/2008JCLI2292.1, 2008.

Laakso, A., Kokkola, H., Partanen, A.-I., Niemeier, U., Timmreck, C., Lehtinen, K. E. J., Hakkarainen, H., and Korhonen, H.: Radiative and climate impacts of a large volcanic eruption during stratospheric sulfur geoengineering, Atmospheric Chemistry and Physics, 16, 305–323, https://doi.org/10.5194/acp-16-305-2016, URL https://www.atmos-chem-phys.net/16/305/2016/, 2016.

Richter, J. H., Tilmes, S., Mills, M. J., Tribbia, J. J., Kravitz, B., MacMartin, D. G., Vitt, F., and Lamarque, J.-F.: Stratospheric Dynamical Response and Ozone Feedbacks in the Presence of SO2 Injections, Journal of Geophysical Research: Atmospheres, pp. n/a–n/a, https://doi.org/10.1002/2017JD026912, URL http://dx.doi.org/10.1002/2017JD026912, 2017JD026912, 2017.

Self, S. and King, A. J.: Petrology and sulfur and chlorine emissions of the 1963 eruption of Gunung Agung, Bali, Indonesia, Bulletin of Volcanology, 58, 263–285, https://doi.org/10.1007/s004450050139, 1996.

Self, S. and Rampino, M. R.: The 1963–1964 eruption of Agung volcano (Bali, Indonesia), Bulletin of Volcanology, 74, 1521–1536, https://doi.org/10.1007/s00445-012-0615-z, 2012.

Stocker, T. F., Dahe, Q., and Plattner, G.-K.: Climate Change 2013: The Physical Science Basis, Working Group I Contribution to the Fifth Assessment Report of the Intergovernmental Panel on Climate Change. Summary for Policymakers (IPCC, 2013), 2013.

Tie, X. and Brasseur, G. P.: The response of stratospheric ozone to volcanic eruptions: Sensitivity to atmospheric chlorine loading, Geophys. Res. Lett., 22, 3035–303, 1995.